# Deconstruction of the beaten Path-Sidestep interaction network provides insights into neuromuscular system development

Hanqing Li[1], Ash Watson[2,3], Agnieszka Olechwier[4], Michael Anaya[1], Siamak K Sorooshyari[5], Dermott P Harnett[2,3], Hyung-Kook (Peter) Lee[1], Jost Vielmetter[1], Mario A Fares[2,6], K Christopher Garcia[7,8], Engin Özkan[4], Juan-Pablo Labrador[2,3]*, Kai Zinn[1]*

[1]Division of Biology and Biological Engineering, California Institute of Technology, Pasadena, United States; [2]Smurfit Institute of Genetics, Trinity College Dublin, Dublin, Ireland; [3]Institute of Neuroscience, Trinity College Dublin, University of Dublin, Dublin, Ireland; [4]Department of Biochemistry and Molecular Biology, University of Chicago, Chicago, United States; [5]Ellipsis Health, San Francisco, United States; [6]Department of Abiotic Stress, Group of Integrative and Systems Biology, Instituto de Biología Molecular y Celular de Plantas (CSIC-Universidad Politécnica de Valencia), Valencia, Spain; [7]Department of Molecular and Cellular Physiology, Howard Hughes Medical Institute, Stanford University School of Medicine, Stanford, United States; [8]Department of Structural Biology, Howard Hughes Medical Institute, Stanford University School of Medicine, Stanford, United States

*For correspondence:
labradoj@tcd.ie (J-PL);
zinnk@caltech.edu (KZ)

**Competing interests:** The authors declare that no competing interests exist.

**Abstract** An 'interactome' screen of all *Drosophila* cell-surface and secreted proteins containing immunoglobulin superfamily (IgSF) domains discovered a network formed by paralogs of Beaten Path (Beat) and Sidestep (Side), a ligand-receptor pair that is central to motor axon guidance. Here we describe a new method for interactome screening, the Bio-Plex Interactome Assay (BPIA), which allows identification of many interactions in a single sample. Using the BPIA, we 'deorphanized' four more members of the Beat-Side network. We confirmed interactions using surface plasmon resonance. The expression patterns of *beat* and *side* genes suggest that Beats are neuronal receptors for Sides expressed on peripheral tissues. *side-VI* is expressed in muscle fibers targeted by the ISNb nerve, as well as at growth cone choice points and synaptic targets for the ISN and TN nerves. *beat-V* genes, encoding Side-VI receptors, are expressed in ISNb and ISN motor neurons.
DOI: https://doi.org/10.7554/eLife.28111.001

## Introduction

Protein-protein interactions (PPIs) control a vast array of processes in metazoans, ranging from signal transduction and gene regulation within cells to signaling between cells *via* cell surface and secreted proteins (CSSPs). The strength of PPIs varies widely, from high-affinity interactions that create stable protein complexes to weak transient interactions (*Nooren and Thornton, 2003*). Defining global PPI patterns ('interactomes') has been the focus of much recent research. Progress has been made in generating high-throughput protein interaction data for a variety of organisms, including *S. cerevisiae* (*Tarassov et al., 2008*), *C. elegans* (*Li et al., 2004*; *Simonis et al., 2009*) and *D. melanogaster*

**eLife digest** Within every organ of the body, cells must be able to recognise and communicate with one another in order to work together to perform a particular role. Each cell has a specific protein on its surface that acts like a molecular identity card, and which can form weak bonds with a complementary protein on another cell. There are thousands of different cell surface proteins, and the interactions between them – known collectively as the interactome – dictate the how cells interact with one another.

Many cell surface proteins are similar across species. Humans and fruit flies, for example, both possess a family of cell surface proteins that contain a region called the Immunoglobulin Superfamily domain. This family can be further divided into subfamilies, two of which are known as "Beats" and "Sides" for short. As the nervous system develops, nerve cells carrying a particular Beat protein interact with nerve or muscle cells carrying a corresponding Side protein. Yet while experiments have matched up many Beats and Sides, the partners of others remain unknown.

Li et al. have now developed a new technique called the Bio-Plex Interactome Assay to rapidly screen for interactions between multiple cell surface proteins in a single sample. Applying the technique to cells from fruit flies revealed new binding partners within the Beats and the Sides. After verifying several of these interactions, Li et al. explored the role of various Beats and Sides in the developing nervous system of fruit fly embryos by mapping the cells that display them on their surfaces.

This increased knowledge of the Beat-Side binding network should provide further insights into how connections form between nerve cells. The new screening technique could also eventually be used to map the cell surface protein interactome in humans. A number of key drugs, including the breast cancer drug Herceptin, target cell surface proteins. Identifying interactions among cell surface proteins could thus provide additional leads for developing new therapies.

DOI: https://doi.org/10.7554/eLife.28111.002

(*Guruharsha et al., 2011*; *Giot et al., 2003*). Methods used to create interactomes include affinity purification/mass spectrometry (AP-MS) and the yeast two-hybrid assay (Y2H).

It is estimated that up to one sixth of human genes encode CSSPs (*Bushell et al., 2008*). CSSPs control signaling from the extracellular milieu to cells and the flow of information between cells. Due to their importance and accessibility, CSSPs are often the targets for therapeutic agents, including humanized monoclonal antibody drugs such as checkpoint inhibitors (*e.g.*, Yervoy and Keytruda), the non-Hodgkin's lymphoma drug Rituxan, and the breast cancer drug Herceptin. However, the bio-chemical properties of many CSSP interactions prevent them from being detected by commonly used techniques employed in high-throughput PPI screens, and CSSPs are underrepresented in global interactomes (*Guruharsha et al., 2011*; *Braun et al., 2009*; *Miller et al., 2005*). There are several reasons for this. First, these proteins are usually glycosylated and have disulfide bonds, so they need to be expressed in the extracellular compartment (*Wright et al., 2010*). CSSP interactions between monomers are also often weak, with $K_D$s in the μM range (*van der Merwe et al., 1994*), making them difficult to capture due to their short half-lives. Lastly, insoluble transmembrane domains on cell surface proteins preclude their purification with standard biochemical techniques, which makes them difficult to study using methods such as AP-MS (*Wright, 2009*).

Despite these difficulties, recent advances have been made in the study of global CSSP interaction patterns. Interactions among cell-surface proteins (CSPs) often occur between clusters of proteins on cell surfaces, and avidity effects (stronger binding due to clustering) can make these cell-cell interactions stable even when monomers bind only weakly. To facilitate detection of interactions among CSSP extracellular domains (ECDs) in vitro, several groups have taken advantage of avidity effects by attaching ECDs to protein multimerization domains and expressing ECD fusions as soluble secreted proteins (*Bushell et al., 2008*; *Wojtowicz et al., 2007*; *Söllner and Wright, 2009*; *Ramani et al., 2012*). These methods have been shown to be effective, allowing detection of inter-actions that otherwise would not have been observable.

Özkan *et al.* scaled up the avidity-based approach, developing a high-throughput ELISA-like screening method, the Extracellular Interactome Assay (ECIA). The ECIA was used to define

interactions among 202 *Drosophila* CSSPs, comprising all CSSPs within three ECD families. These were the immunoglobulin superfamily (IgSF), fibronectin type III (FNIII) and leucine-rich repeat (LRR) families. The ECIA utilized dimers as 'bait' and pentamers as 'prey'. It detected 106 interactions, 83 of which were previously unknown (*Özkan et al., 2013*).

The most striking finding from the ECIA interactome was that a subfamily of 21 2-IgSF domain CSPs, the Dprs, selectively interacts with a subfamily of 9 3-IgSF domain CSPs, the DIPs, forming a network called the 'Dpr-ome' (*Özkan et al., 2013*). Each Dpr and DIP that has been examined is expressed by a small and unique subset of neurons at each stage of development. One Dpr-DIP pair is required for normal synaptogenesis and influences neuronal cell fate. In the pupal optic lobe, neurons expressing a Dpr are often presynaptic to neurons expressing a DIP to which that Dpr binds in vitro (*Carrillo et al., 2015*; *Tan et al., 2015*). The Dpr-ome initially defined by the global interactome contained several 'orphans', proteins with no binding partner (*Özkan et al., 2013*). By expressing new versions of Dprs and DIPs, including chimeras, and using these to conduct a 'mini-interactome' analysis of the Dpr-ome, we were able to find partners for all but one orphan. That protein, Dpr18, has changes to conserved binding interface residues and may lack the capacity to bind to any DIPs (*Carrillo et al., 2015*).

The ECIA also identified a second IgSF network, formed among members of the Beaten Path (Beat) and Sidestep (Side) protein subfamilies. Beat-Ia and Side were identified by genetic screens for motor axon defects, and were later shown to have a ligand-receptor relationship. They control the projection of motor axons to muscle targets (*Fambrough and Goodman, 1996*; *Sink et al., 2001*; *de Jong et al., 2005*; *Siebert et al., 2009*). Beat-Ia is expressed on motor axons, where it binds to Side, which is expressed on muscles. This binding causes motor axons to decrease their adhesion to each other, allowing them to leave their bundles and turn onto the muscle fibers. *beat-Ia* and *side* have strong motor axon phenotypes. In the absence of either protein, motor axons often remain in their fascicles and never leave to arborize on their target muscles (*Siebert et al., 2009*; *Aberle, 2009*).

The ECIA detected the known Beat-Ia::Side interaction, and also uncovered other interactions between members of the Beat and Side subfamilies (*Özkan et al., 2013*). Seven of the 14 Beats were found to bind to four of the eight Sides. The remaining Beats and Sides were still orphans with no binding partners in the other subfamily. The functions of the newly defined interactions between Beats and Sides were unknown. Most *beat* genes are expressed in embryonic neurons. Some Beats were genetically characterized using deletion mutations and RNAi, but loss of these Beats did not cause strong motor axon phenotypes (*Pipes et al., 2001*). None of the other Side subfamily members had been examined.

This paper describes the development of a new method for interactome screening, which we call the BPIA (Bio-Plex Interactome Assay). This method uses the 'Bio-Plex' system, which employs Luminex xMAP technology (*Houser, 2012*). Our method detects binding of a prey protein to many bait proteins, each conjugated to a bead of a different color, in each assay well. For the ECIA, the number of assays required for the interactome screen was the square of the number of proteins examined, while with the Bio-Plex the number of assays could be equal to the number of proteins. In principle, then, the Bio-Plex might greatly speed up interactome screening, and might also be more sensitive, since the available signal-to-background ratio is much greater for the Bio-Plex than for the ECIA. As a test of the method, we used a Bio-Plex 200 to do a mini-interactome screen of the Beat-Side network. Based on the the fact that the Dprs and DIPs that were initially orphans (*Özkan et al., 2013*) were later shown to have binding partners (*Carrillo et al., 2015*), we hypothesized that some of the orphan Beats should have Side partners, and vice versa. Consistent with this hypothesis, we were able to deorphanize two more Beats and two Sides using the BPIA.

To further our understanding of Beat and Side function during embryonic development, we characterized expression of several Beats and Sides, focusing primarily on Side-VI and the three Beat-Vs, which were the strongest interactors in both the ECIA and BPIA screens. The three *beat-Vs* exhibit differential expression in identified motor neurons, while *side-VI* is expressed at motor axon choice points and in a subset of target muscle fibers.

## Results

### Side belongs to an IgSF subfamily whose origin predates drosophilid speciation

Gene duplication, a key phenomenon in the expansion of gene families, provides opportunity for the fine-tuning or innovation of protein interactions and functions (*Ohno, 1970*). The duplication of genes encoding receptors or ligands that have multiple binding partners can lead to partitioning of the interactions among the paralogs. Relaxed constraints due to redundancy between duplicated genes can result in the exploration of new functions. In these ways, members of ligand and receptor families can establish a complex interaction network in which each binding pair has a distinct expression pattern and function.

The Beat IgSF subfamily was previously characterized in *Drosophila melanogaster* (*Pipes et al., 2001*). Here we show that orthologs of each Beat are found in most of the other 12 sequenced Drosophilid species (*Figure 1A*). Beats have two IgSF domains, and there are both secreted and membrane-bound isoforms. There are 14 Beat proteins, divided into seven clusters based on their phylogenetic relationships. Beats that are most closely related to each other (*e.g.,* Beats Ia, Ib, and Ic) are encoded by clustered genes and denoted by a Roman numeral followed by a letter. While divergence rates within the *beat* family phylogeny are highly asymmetric following the earliest duplications, groups of *beats* within each of the clusters of paralogs have similar divergence levels (*e.g.,* Beat-IIa and Beat-IIb present similar rates of evolution). Beats encoded within clusters also have similar (but not identical) embryonic expression patterns (*Pipes et al., 2001*). These observations suggest that *beats* have undergone two levels of specialization: functional specialization after duplication and emergence of the seven major Beat branches, followed by individuation of expression patterns and binding specificities for members of the four subclusters (I, II, III, V).

Proteins in the Side subfamily are all transmembrane proteins, and none are encoded by clustered genes (*Sink et al., 2001*; *Aberle, 2009*). An exhaustive analysis using SMART (*Schultz et al., 1998*), HMMER (*Eddy, 2011*) and DOUT-finder (*Novatchkova et al., 2006*) to identify outlier homologs of structural domains reveals that the Side family of paralogs has an invariant ECD architecture composed of five IgSF domains followed by an FNIII domain (*Figure 1B*).

In addition to the protein domain-based composition, phylogenetic inferences provide evidence for a cohesive subfamily of Side proteins (*Figure 1C*). We refer to Sidestep as Side, and have designated names for the other seven Side paralogs based on their evolutionary distance from Side. All Side paralogs seem to present similar or comparable levels of inter-species (intra-paralog) divergence, indicating that different Side paralogs have undergone similar selective constraints. The presence of Side paralogs in most of the 12 sequenced Drosophilids, and the presence of orthologs of some of these paralogs in the mosquito *Anopheles gambiae,* clearly indicates the origin of the Side family through successive duplication events that pre-dated Drosophilid speciation.

We could not identify orthologs in all 12 Drosophilids for all Sides, likely due to incomplete genomic sequence (see Materials and methods). The most likely scenario given our phylogenetic trees is that the ancestral Side subfamily gene duplicated successively, followed by a rapid sequence and functional divergence predating Drosophilid speciation. Indeed, rooted trees for the Side family show a dynamic history of gene duplication and divergence, with asymmetric clusters of duplicates resulting from faster evolution of one gene copy compared to its sister, indicating possible functional divergence and specialization after gene duplication. Our rooted phylogeny of the Side paralogs differs from a previous unrooted one (*Aberle, 2009*). The low bootstrap support values (p<60%) for some of the internal tree branches indicate rapid successive duplication events. The long branches post-dating duplications but predating speciation support enormous divergence between the duplicates at the sequence level, followed by strong purifying selection after speciation.

### Development of the BPIA, a new high-throughput CSSP interaction assay

To attempt to deorphanize more members of the Beat and Side superfamilies, and to develop new assays that might eventually streamline the process of creating global interactomes, we investigated technologies that have the potential to provide higher throughput and greater sensitivity. The Bio-Rad Bio-Plex system is based on the principles of flow cytometry and can be used for a variety of

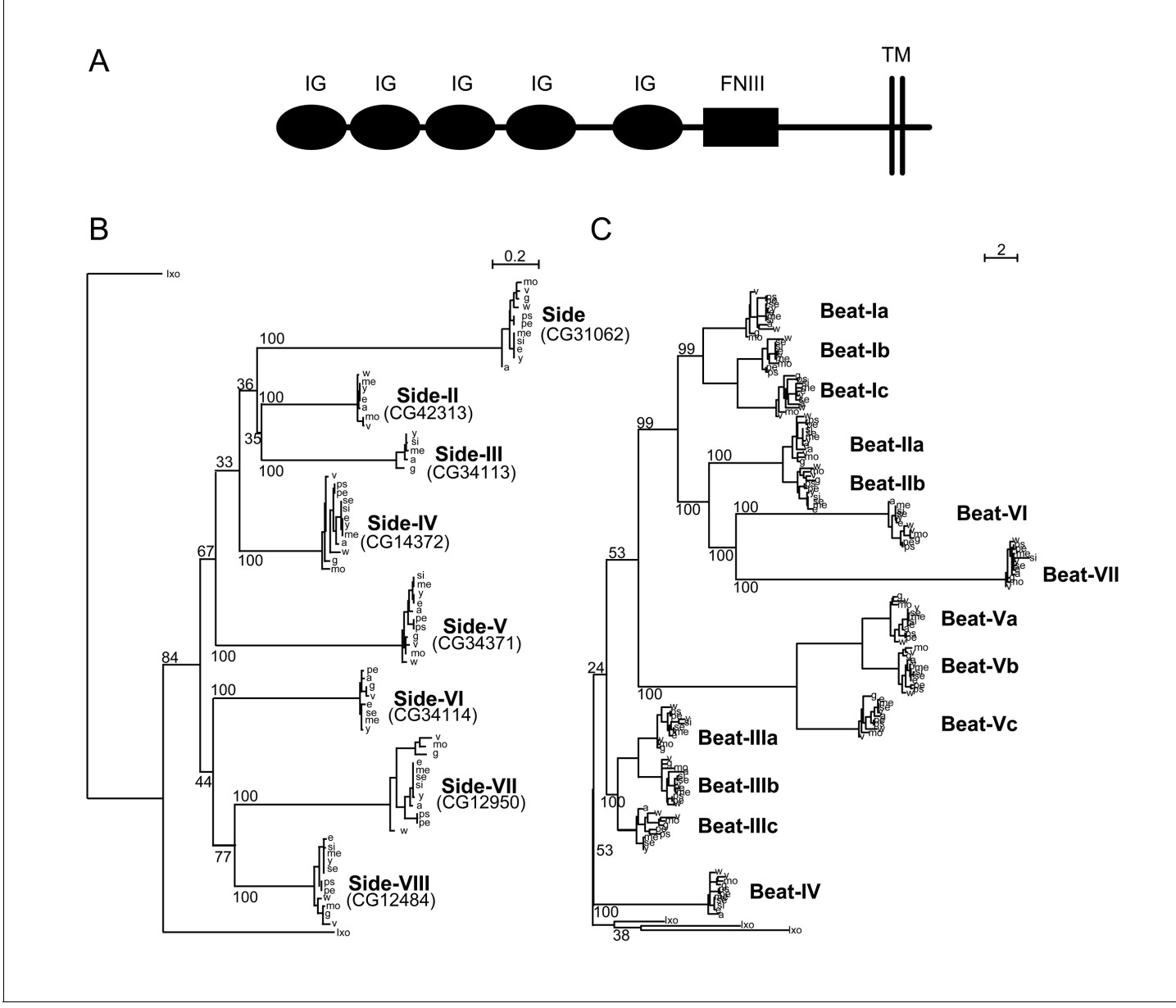

**Figure 1.** Phylogenetic analysis of Beaten Path and Sidestep paralogs. (A) Phylogeny of the Beat family of receptors rooted against the tick *Ixoides scapularis* (Ixo) Beats. Beat-VII and Beat-VI share a more recent ancestor than previously described. (B) Extracellular architecture of the Side subfamily. Detailed ClustalW alignment of individual domains and conservation are in *Figure 1—figure supplement 1*. (IG, immunoglobulin superfamily; FNIII, Fibronectin type III; TM, transmembrane domain). (C) Phylogeny of the Side family of related proteins rooted against similar IgSF proteins predicted in the tick, *Ixoides scapularis* (Ixo) that form a distinct outgroup. Names are assigned to the paralogs on the basis of their evolutionary distance from Sidestep and their CG Flybase identifiers are in parentheses.

DOI: https://doi.org/10.7554/eLife.28111.003

The following figure supplement is available for figure 1:

**Figure supplement 1.** Sequence alignments of extracellular domains in the Side family.
DOI: https://doi.org/10.7554/eLife.28111.004

high-throughput, multiplexed assays. It uses magnetic polystyrene beads impregnated with different ratios of fluorescent dyes (each variant is called a 'bead region'), rendering them spectrally distinct when excited by a laser. The Bio-Plex 200 used in our experiments employs two dyes and has 100 different bead regions, allowing for the simultaneous analysis of up to 100 distinct bead-bound analytes, while the Bio-Plex 3D has 500 bead regions. The beads can be conjugated to lysine residues

on proteins through carboxyl groups on their surfaces. Protein-conjugated bead regions are mixed and incubated with soluble proteins. Binding between soluble and bead-bound proteins can be detected using phycoerythrin (PE)-coupled secondary antibodies or other fluorescent reagents. Beads flow through the machine in single file, and are interrogated by two lasers: one to discern the identity of the bead region, and the other to detect the PE signal, representing the amount of bound binding protein. This assay has a high signal-to-background ratio, because strong binding of the bead-bound analyte can generate readings of >20,000, *vs.* <100 for unconjugated beads.

Luminex xMAP technology has been used by other groups to assay interactions between proteins (*Blazer et al., 2001*; *Rimmele et al., 2010*; *Blazer et al., 2011*). For example, Blazer et al. used avidin-coupled bead regions to capture Regulator of G protein Signaling (RGS) proteins. These were then incubated with fluorescently labeled $G_{\alpha o}$ to measure RGS-G protein interactions and identify compounds that could inhibit these interactions.

In developing a Bio-Plex-based assay for CSSP interactions, we took advantage of avidity, utilizing the same dimer and pentamer constructs employed for the ECIA, but reversing the bait and prey roles (*Özkan et al., 2013*). Bait proteins were alkaline phosphatase (AP) fusion proteins of ECDs pentamerized using a sequence from cartilage oligomeric matrix protein (COMP) (*Bushell et al., 2008*; *Voulgaraki et al., 2005*). Prey constructs were fusions of ECDs to human Fc, which is a dimer. Preys contain a C-terminal V5 epitope tag, so a V5 antibody was used to detect binding, followed by a secondary antibody conjugated to PE.

We developed an affinity-capture method to attach bait proteins to the beads that avoided the necessity to purify baits (*Figure 2A*). To accomplish this, we added a sequence encoding an Avitag, a 15 amino acid sequence that is recognized by the enzyme biotin ligase (BirA), at the C-terminal end of each bait protein coding region. BirA adds one biotin molecule to the tag (*Ashraf et al., 2004*; *Sung et al., 2011*; *Wang et al., 2013*). To perform in vivo biotinylation, we co-transfected the bait constructs with an endoplasmic reticulum (ER)-localized BirA construct optimized for expression in S2 cells (*Tykvart et al., 2012*). To capture bait proteins, we coupled each bead region to streptavidin, and incubated each with media containing a different biotinylated bait protein, thus bypassing the purification step. Each Fc-tagged prey protein was also expressed in S2 cells, and purified with Ni-NTA resin. The bait-coated beads were then mixed and aliquoted and a different Fc prey protein added to each tube. The reactions were then washed and incubated with anti-V5 antibody, followed by PE-conjugated secondary antibody, before being transferred to a 96-well plate and read with the Bio-Plex. This is the BPIA assay.

The biotin-streptavidin interaction is one of the strongest non-covalent interactions known in nature, with a $K_D$ on the order of $\sim10^{-14}$ mol/L (*Hendrickson et al., 1989*), so we expected that bait proteins should not be able to 'jump' to other beads after bead regions are mixed. To test this, we incubated multiple bead regions with bound baits overnight together with a streptavidin-coupled bead region lacking a bait. We observed no jumping of baits to the bead region without a bait (see Materials and methods for details).

## Using the BPIA to assay interactions between Beats and Sides

We performed a Beat-Side mini-interactome for the 22 Beat and Side subfamily members plus the IgSF protein CG17839, which binds to Side-VII (*Özkan et al., 2013*), using the Bio-Plex system. The purposes of this experiment were to: (1) demonstrate that we could screen for binding of all baits to a prey within a single well, and therefore that we could do the entire assay with 23 wells plus controls (*vs.* the 529 wells that would have been required by the original ECIA), and, (2) determine if we could observe all of the interactions found by the ECIA. We also hoped that the BPIA might be more sensitive than the ECIA, due to its high signal-to-background ratio, and therefore might uncover previously unknown interactions.

23 bait constructs were co-transfected with the BirA plasmid in S2 cells, and AP bait protein-containing media harvested. The bait proteins were then captured directly from media using streptavidin-coupled beads, and the beads mixed together. As prey, 23 different His-tagged ECD-Fc constructs were transfected into S2 cells, and the fusion proteins purified with Ni-NTA resin. We analyzed each potential interaction pair in both orientations, with protein A as bait and protein B as prey, and vice versa. The pooled beads were incubated with a prey protein overnight, washed, and then incubated with anti-V5 antibody followed by PE-conjugated secondary antibody, transferred to

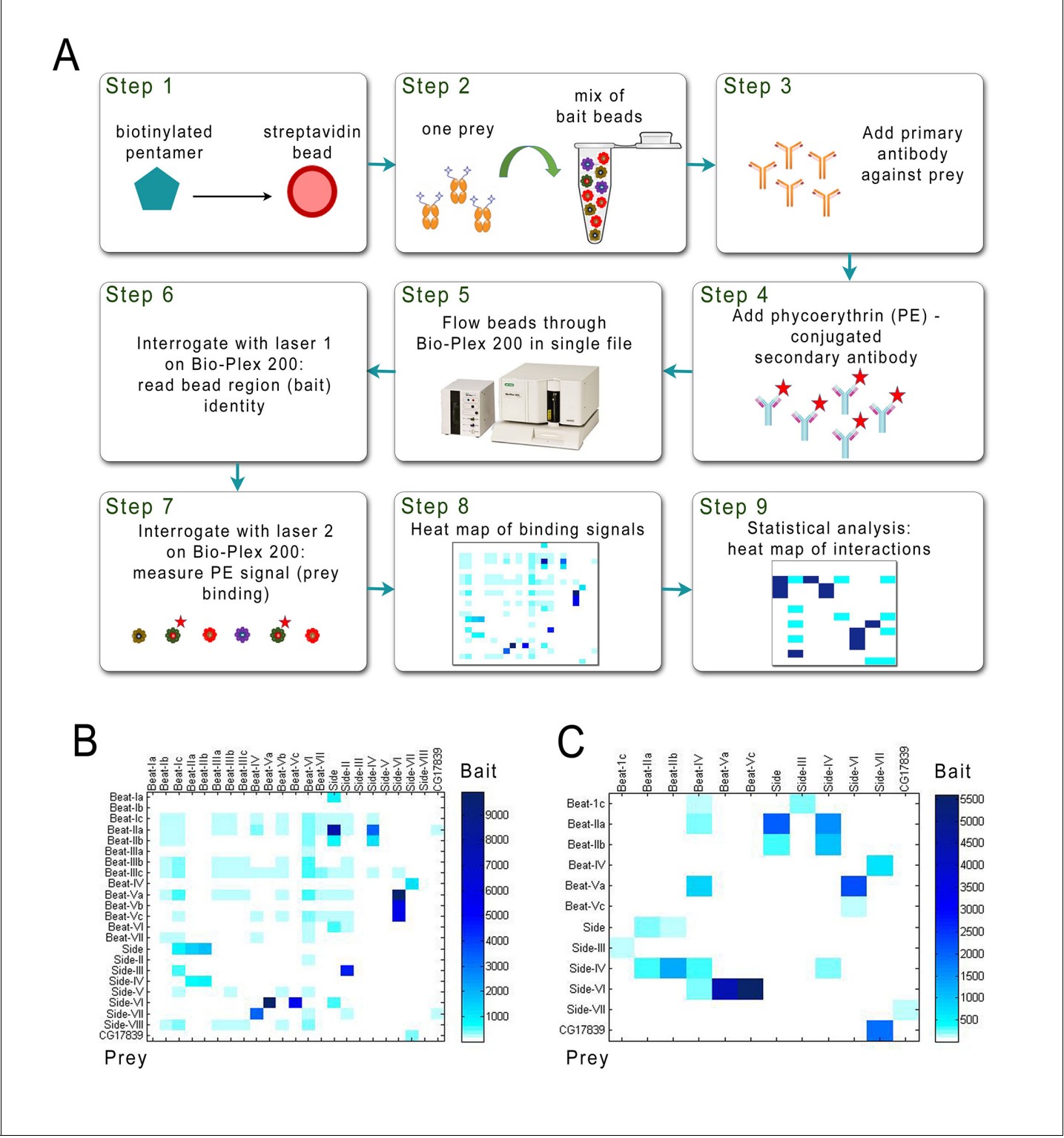

**Figure 2.** Schematic of the BPIA and heat maps of Beat/Side interactions. (**A**) Each biotinylated AP bait was captured from media with streptavidin-coupled beads from a particular bead region (Step1), and all bead regions were pooled and aliquoted. A different Fc prey was added to each aliquot and incubated overnight (Step 2). Primary antibody was added to the reactions the next day (Step 3), followed by phycoerythrin (PE)-conjugated secondary antibody (Step 4). The reactions were then transferred to a 96-well plate. In the Bio-Plex 200 machine, beads are aspirated in single file from the wells (Step 5) and interrogated by two lasers. The first laser reads the color ('region') for each bead, and therefore identifies which bait is being analyzed (Step 6). The second laser reads the PE signal, and the strength of this signal indicates how much prey is bound to each bead region (Step 7).
*Figure 2 continued on next page*

*Figure 2 continued*

These data are used to produce a heat map of raw interaction signals (Step 8). After statistical analysis, likely receptor-ligand interactions are defined (Step 9). (B) 23 × 23 matrix of raw interaction signals between Beats and Sides performed using purified prey. (C) 12 × 12 matrix of raw interaction signals between Beats and Sides using unpurified prey. All interactions seen with purified prey were also detected using unpurified prey, except for Beat-Ic::Side.

DOI: https://doi.org/10.7554/eLife.28111.005

a 96-well plate, and analyzed with the Bio-Plex. *Figure 2B* shows a heat map of the raw interaction signals for the 23 × 23 matrix.

Direct protein capture with streptavidin-coupled beads enabled us to bypass protein purification for bait proteins. We were interested to see if the assay could also be performed using unpurified prey proteins, which would further reduce the workload involved. To test this, we performed the BPIA using a subset of the Beat and Side subfamily proteins. Bait proteins were expressed and captured as described above. The prey proteins were expressed in S2 cells grown in Sf-900 III, a serum-free media optimized for protein expression in insect cells. We chose serum-free media due to the fact that the high concentration of extraneous proteins present in normal (serum-containing) S2 media lowers the signal to background ratio of the assay (data not shown). Using this method, we were able to find all of the interactions seen with purified prey, except for Beat-Ic::Side (perhaps due to low expression of Beat-Ic-AP bait in this experiment) (*Figure 2C*). These results show that the BPIA is also compatible with the use of unpurified prey proteins.

## New interactions identified using the BPIA

To analyze the Beat-Side mini-interactome, we utilized methods based on those of Özkan et al, who used a Z score system to classify interactions (*Özkan et al., 2013*). They used a cutoff to eliminate outliers, but we could not do this because our data set is much smaller and a large fraction of the proteins interact (see Materials and methods). Accordingly, to process our data, we utilized bootstrapping of the median for each row and each column. Briefly, for every row and column, 23 numbers were chosen randomly with replacement, and the median calculated. After $n$ cycles, a histogram of the median was generated and the mean and SD for that row or column calculated. A $Z$ score was then calculated for each number in the row or column based on the generated mean and SD. In this manner, for each number in the matrix, two different $Z$ scores were generated (one based on row, the other on column). The two $Z$ scores were then averaged.

Each Beat-Side protein pair appears twice in the matrix, since each protein is used as both bait and prey. The two averaged $Z$-scores for each pair represent interactions in opposite orientations. It was expected that these values would be discrepant, as in the ECIA, due to differences in protein expression, binding geometries, and other factors. To incorporate interactions in both directions into the analysis, we calculated the geometric mean (square root of the product) of the two $Z$ scores (note that geometric mean can only be calculated if both scores are >0). If the geometric mean was greater than five, the Beat-Side pair was scored as a genuine interactor.

*Figure 3A* graphically displays these results. It is a quantized heat map generated from the geometric means of $Z$ scores for each protein pair. Each $Z$ score was assigned to one of three categories: high (dark blue), mid (light blue), and low (white), which were determined using cutoffs of 80% and 90%. These cutoffs were chosen so that the 'high' $Z$-score category corresponded to the pairs we scored as genuine interactors, having a geometric mean of $Z$-scores that was >5. For comparison, *Figure 3B* shows a heat map based on the ECIA data (*Özkan et al., 2013*). All of the hits found with the ECIA were in the high category, with the exception of Beat-Ia::Side. This is likely due to poor expression of Beat-Ia-AP bait, resulting in one of the $Z$ scores being zero. Note that in *Figure 2B* a Beat-Ia::Side interaction is detected when Beat-Ia is the Fc prey and Side is the AP bait. Beat-VI:: Side-II, Beat-Ic::Side-III, and Beat-Ic::Side are in the high category in our heat map, and we concluded that these are new interacting pairs. The 'revised' Beat-Side subfamily interaction network is shown in *Figure 3C*, with new interactions identified by the BPIA indicated by red lines.

Interestingly, we observe a 'phylogenetic mirroring' between Sides and Beats when we compare the network diagram (*Figure 3C*) with the evolutionary trees of *Figure 1*. Sides closer to the root of the tree tend interact with Beats also close to the root of the tree and vice versa. Side members

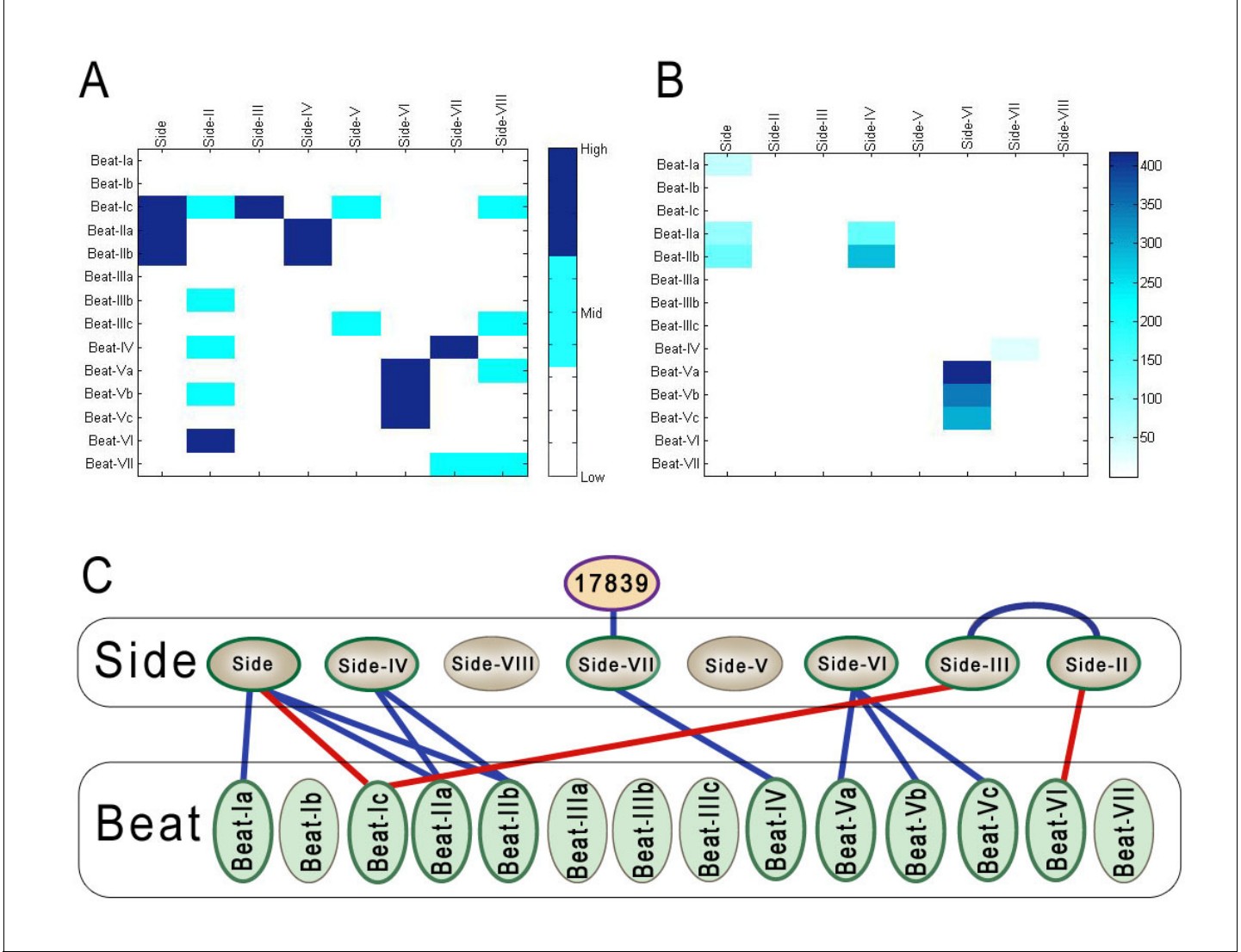

**Figure 3.** Comparison of BPIA and ECIA data, and the revised Beat-Side network map. (**A**) A 14 × 8 quantized heat map of interactions between Beats and Sides determined using the BPIA. This heat map was generated using the geometric means of *Z* scores for interactions between Beats and Sides calculated using the numbers in *Figure 2B*. Each number in the matrix was then assigned to one of three values: low, mid and high. These values were calculated using cutoffs of 80% and 90%. (**B**) 14 × 8 heat map of geometric means of Beat/Side interactions using ECIA data from Özkan et al. (**C**) Updated network of Beat-Side interactions. Three new interactions were discovered with the BPIA: Beat-VI::Side-II, Beat-Ic::Side, and Beat-Ic::Side-III.
DOI: https://doi.org/10.7554/eLife.28111.006

separated by more branchpoints from the root of the tree may have become functionally specialized to interact with the more recently duplicated Beats.

## Measuring the affinities of Beat/Side interactions

To confirm the interactions between Beat and Side subfamily proteins observed in the ECIA and BPIA and determine their affinities, we used surface plasmon resonance (SPR). We purified monomeric ECDs from proteins expressed using the baculovirus system. For the Beat-V::Side-VI interactions found in the ECIA, we flowed Beat-V ECDs over the surface of Biacore chips layered with Side-VI to determine their binding affinities and the kinetics of the interactions. Binding data show that dissociation kinetics are too fast to measure ($k_{off} \geq 0.5$ s$^{-1}$). Therefore, SPR responses are only fitted at equilibrium to a binding isotherm and their fit is indicative of specific interaction. Affinities ($K_D$s) for binding of the three Beat-Vs to Side-VI are in the μM range (0.76 μM, 2.3 μM and 9.4 μM for

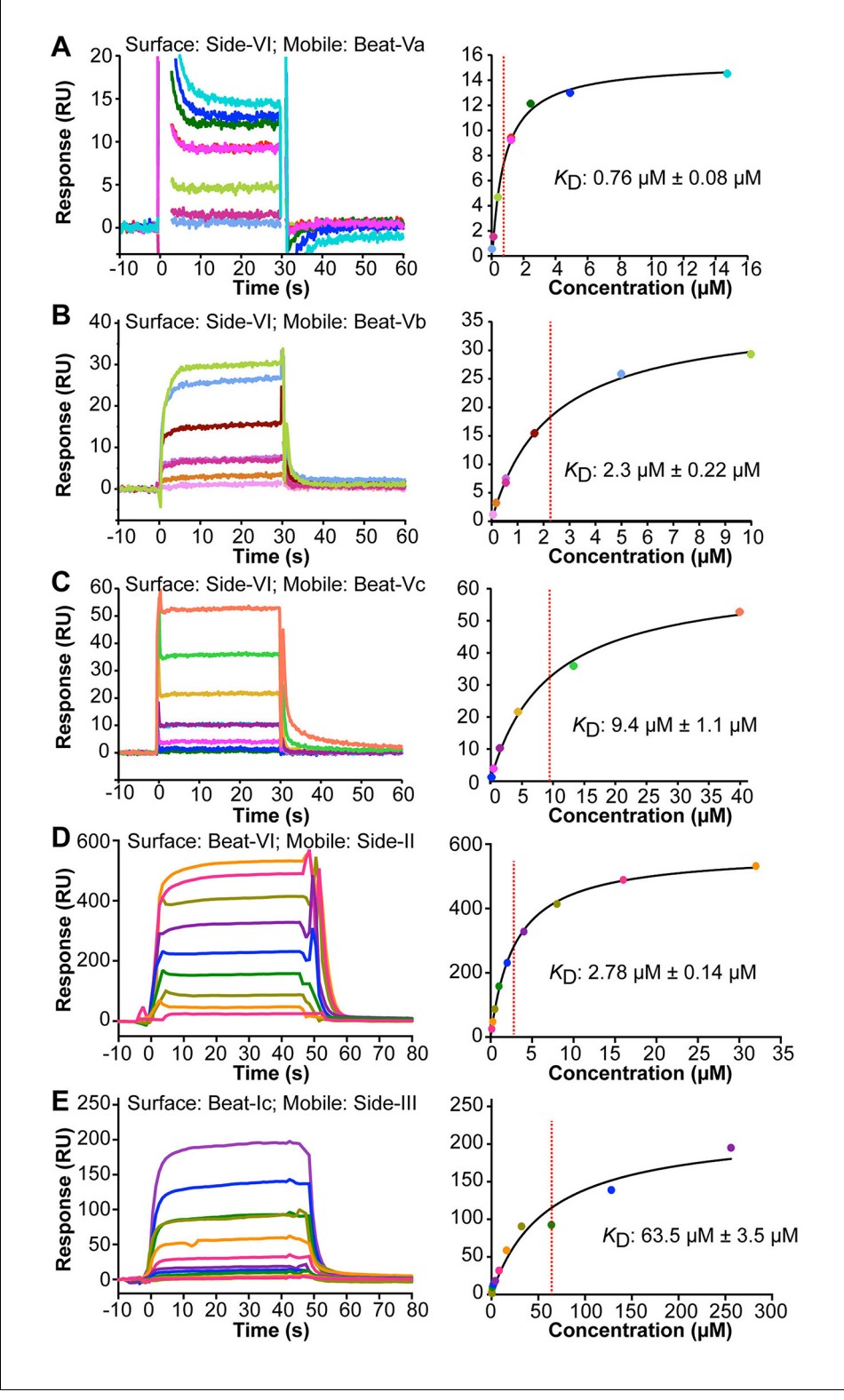

**Figure 4.** Surface Plasmon Resonance sensorgrams (left panels) and binding isotherms (right panels) for five Beat–Side complexes. Equilibrium binding responses are fit to Langmuir isotherms to calculate dissociation constants ($K_D$). Each color in the sensorgrams represents the concentration of the analyte in mobile phase. Zero time-point indicates time of analyte injection. The color scheme from the sensorgrams is preserved in the binding isotherms. (**A–C**) Interactions of Side-VI with the Beat-V family of receptors. Side-VI was captured on a Biacore SA chip, and

*Figure 4 continued on next page*

*Figure 4 continued*
titration series of Beat-Va (**A**), Beat-Vb (**B**), and Beat-Vc (**C**) were flowed over the SA chip. The ±errors represent standard error from the fitting of one titration series. (**D and E**) Interactions of Beat-VI with Side-II (**D**), and Beat-Ic with Side-III (**E**). Beat-VI and Beat-Ic were captured on a Biacore SA chip, and the Sides were flowed over the chip. The ±errors represent standard error of the mean for $K_D$ from three titration series.
DOI: https://doi.org/10.7554/eLife.28111.007

Beat-Va, Beat-Vb and Beat-Vc, respectively; *Figure 4A–C*). These dissociation constants are in the same range as those we have previously described for interactions between Beat-Ia and Side (*Özkan et al., 2013*) and are typical for interactions of cell adhesion molecules (*van der Merwe et al., 1994*).

We also examined the three new Beat/Side interactions discovered with the BPIA: Beat-VI::Side-II, Beat-Ic::Side-III, and Beat-Ic::Side. To verify these interactions, we also measured them using SPR. Beat-VI and Beat-Ic ECDs were captured on Biacore chips and Side-II and Side-III were flowed over the chips. Interactions were observed between Beat-VI and Side-II ($K_D$: 2.78 µM) and Beat-Ic and Side-III ($K_D$: 63.5 µM) (*Figure 4D,E*). Binding was also observed between Beat-Ic and Side, although Side ECD precipitation precluded the collection of a titration series.

## Binding of Side-VI to Beat-Vs on embryos and cells

The strongest interactions observed in both the ECIA and BPIA were those between Beat-Vs and Side-VI (*Figures 2B* and *3*). Interestingly, with both assay the observed signals were related to the measured $K_D$s for these interactions, with Beat-Va (tightest binder)>Beat-Vb>Beat Vc. To further characterize these interactions, we expressed Beat-Va, Beat-Vb, and Beat-Ia on the surfaces of S2 cells, and evaluated their binding to Side-VI-AP. We observed binding for both Beat-Va and Vb, but not for Beat-Ia (*Figure 5—figure supplement 1*). We then determined if Side-VI could bind to Beat-Vb in embryos by live staining with Side-VI-AP (*Özkan et al., 2013*; *Lee et al., 2009*; *Fox and Zinn, 2005*). To do this experiment, we expressed Beat-Vb from a UAS construct using a strong pan-cellular GAL4 driver, Tub-GAL4. *Figure 5* shows that Side-VI-AP strongly stains muscle fibers in Tub >Beat Vb embryos, but not in wild-type embryos. In wild-type embryos, punctate Side-VI-AP staining is observed on motor axons (*Figure 5—figure supplement 2*), consistent with the fact that Beat-Vs are expressed by motor neurons (see below).

## Expression patterns of *side* subfamily genes

The expression pattern of Side protein is dynamic in space and time. Its expression pattern changes as motor axons grow toward their targets, so that Side marks the cells over which growth cones travel during each stage of development (*Sink et al., 2001*; *Siebert et al., 2009*). At stage 12 it is expressed in cells in a belt-like pattern within the CNS and slightly later in a cluster of cells with a tri-angular pattern that are contacted by pioneer motor axons in the intersegmental nerve (ISN) on their way to the dorsal muscle field. At later stages Side is expressed in sensory afferents, where it is downregulated following contact with Beat-Ia expressing motor axon growth cones, and Side subsequently appears on the muscle fibers. Thus, Side labels substrates followed by ISN axons at each stage of their growth toward their muscle targets (*Siebert et al., 2009*). We reasoned that expression of *side* paralogs at guideposts or choice points along these nerve tracts would be an indicator of other Beat-Side interactions that might be important for motor axon guidance. Therefore, we examined the embryonic expression of *side-II, side-III, side-IV, side-VI, side-VII* and *side-VIII* by fluorescent in situ mRNA hybridization (FISH) and labeled all motor axons with the anti-fasciclin-II antibody mAb 1D4, to assess the coordinates of *side* paralog expression relative to motor axon trajectories.

All of these side genes, with the exception of *side-VIII*, are expressed in peripheral tissues traversed by motor and sensory axons. *side-II* is transcribed broadly in the CNS and to a lesser extent in the developing musculature at stage 15 (*Figure 6A*). *side-III* is initially expressed at high levels in the mesoderm and muscle primordia and broad transcription in the CNS increases as embryonic development progresses. By stage 14 peripheral *side-III* expression is strongest in the developing trachea and in stripes in the ectoderm along the parasegmental furrows (*Figure 6B,N*). The tracheal

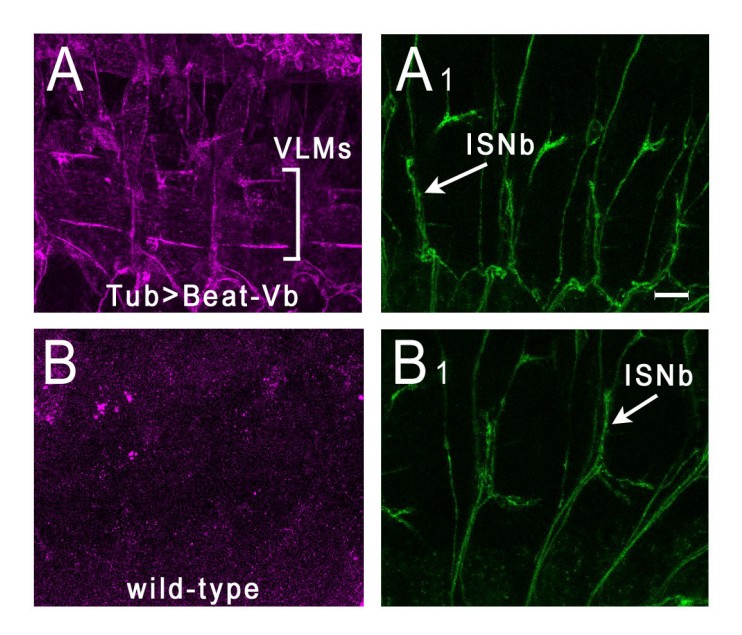

**Figure 5.** Side-VI-AP binds to Beat-Vb in live embryos. Stage 16 embryos were live-dissected and stained with Side-VI-AP and 1D4. (A), (A₁) Tub-GAL4 >UAS Beat-Vb embryos stained with Side-VI-AP (A) and mAb 1D4 (A₁). Strong Side-VI-AP staining of muscle fibers is observed. The ventrolateral muscle field is indicated by brackets. Note that the lateral muscle fibers (muscles 21–24) above the VLMs are outlined by Side-VI-AP staining. (B), (B₁) WT embryos stained with Side-VI-AP (B) and 1D4 (B₁). There is no staining of muscle fibers above background. Scale bar, 12 μm.

DOI: https://doi.org/10.7554/eLife.28111.008
The following figure supplements are available for figure 5:

**Figure supplement 1.** The Side-VI ectodomain binds to Beat-Va and Beat-Vb expressed on the surfaces of S2 cells.
DOI: https://doi.org/10.7554/eLife.28111.009
**Figure supplement 2.** The Side-VI ectodomain binds to motor axons in live-dissected embryos.
DOI: https://doi.org/10.7554/eLife.28111.010

branches are known intermediate targets of the ISN and sensory axons (*Younossi-Hartenstein and Hartenstein, 1993*; *Harris and Whitington, 2001*). *side-IV* is expressed in ventral muscle precursors at stage 12, and by stage 14 it is localized to ventral muscles (muscles 15, 16, and 17) and to lateral muscles 5 and 8 (*Figure 6C,D,N*). *side-VII* shows broad expression in the CNS at stage 14, and is also detected in the dorsal tracheal trunk (*Figure 6K,N*). *side-VIII* expression is quite different from these other *side* genes. There is no detectable expression outside the CNS (data not shown), and at stage 16 expression is restricted to a subset of CNS neurons, including the RP1, 3, 4, and five motor neurons and the pCC interneuron (*Figure 6L,M*).

The pattern of expression of *side-VI* is of particular interest, because it is expressed at key choice points for motor axons in the ISN, intersegmental nerve b (ISNb) nerves, and transverse (TN) nerves. It is broadly expressed in the CNS (*Figure 6E,I*). At stages 15 and 16 it is expressed in subsets of muscle fibers, including ventrolateral muscles 12 and 13, which are the targets of the RP4 and RP5 ISNb motor axons (*Figure 6J,N*). *side-VI* is also transcribed in cells whose surfaces are explored by the ISN tip, such as the dorsal cluster of Lim3-positive sensory neurons that fasciculate with the ISN (*Figure 6F,N*), and in a 'persistent Twist expressing cell' (PT cell) which coincides with the first branch point of the ISN within the dorsal musculature (*Bate et al., 1991*) (*Figure 6G,N*). *side-VI* is also expressed in certain targets of the TN that are known to be essential for its guidance, including the lateral bi-dendritic neuron (LBD) (*Figure 6H,N*) and the dorsal median cell (DMC) (*Chiang et al., 1994*; *Gorczyca et al., 1994*) (*Figure 6I,N*).

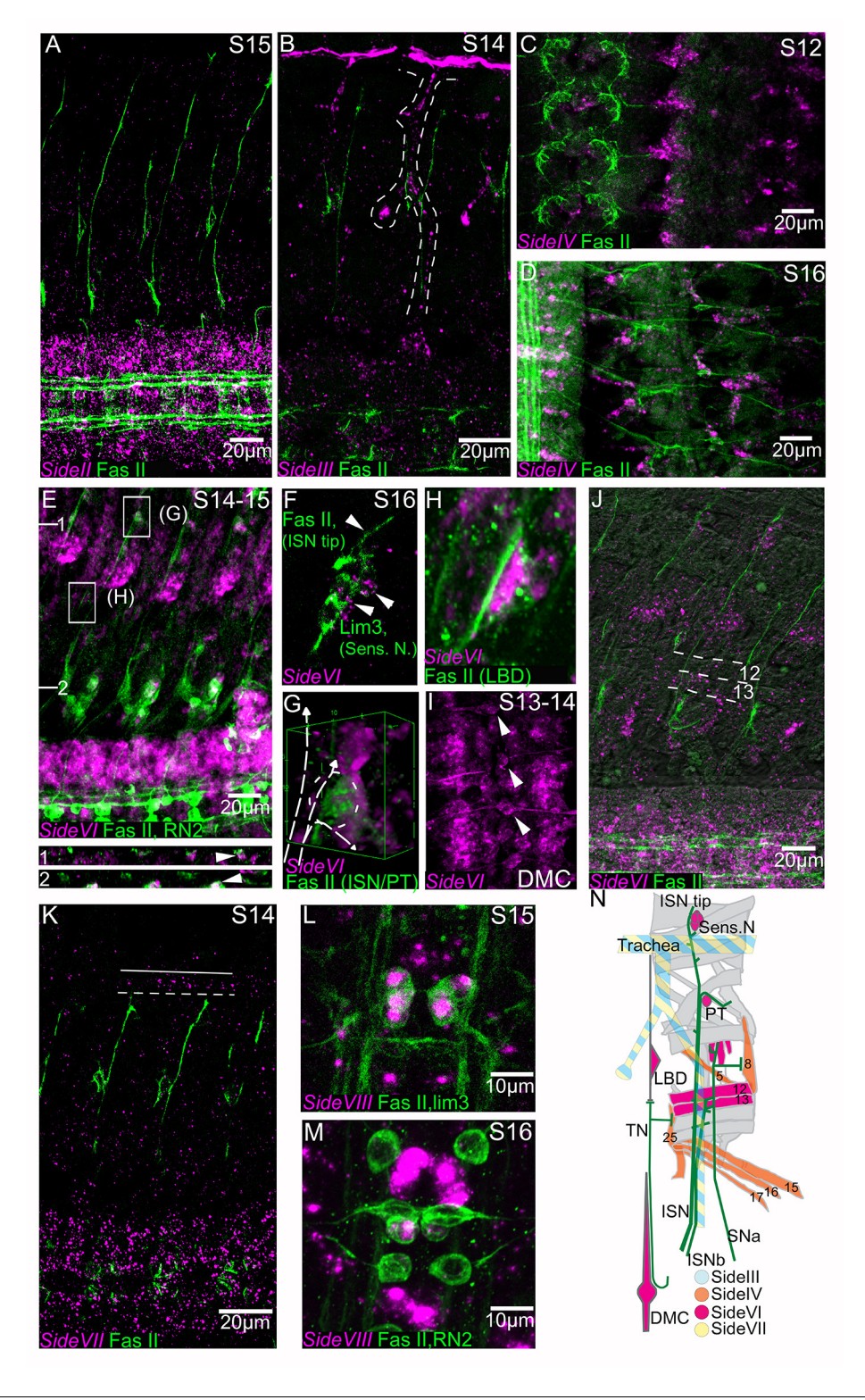

**Figure 6.** Embryonic expression patterns of *side* genes. Fluorescent in situ hybridization (magenta) of *side-II, side-III, side-VI, side-VII and side-VIII* genes in fillet preparations. All preparations are co-stained with anti-Fasciclin II antibody (1D4) to reveal all motor nerves (green). (**A**) *side-II* is predominantly expressed in the CNS, where it has an increasingly broad expression pattern as development progresses. (**B**) *side-III* expression pattern in a stage 14 embryo in the developing trachea (dashed line). (**C**) *side-IV* expression pattern in a stage 12 embryo in ventral

*Figure 6 continued*

muscle precursors. (**D**) At stage 16 expression of *side-IV* is localized to the ventral oblique muscles (muscles 15, 16, 17), the ventral transverse (25), the lateral oblique (5) and the segment border (8) muscles. (**E**) Expression of *side-VI* at stage 14–15 co-stained for RN2-Gal4 > UAS tau-LacZ. *side-VI* is broadly expressed in the CNS and in specific tissues in the periphery. XZ sections are indicated and represented underneath the main panel and magnifications of selected areas (**G, H**) are presented in individual panels. Orthogonal views show a cross section of a dorsal set of sensory neurons (1) and the junction of the ISN at its first branch, FB (2). The location of the ISN is marked with an arrowhead. (**F**) In a stage 16 embryo the ISN tip explores a group of *side-VI* expressing dorsal sensory neurons. (**G**) 3D projection of the ISN FB region where *side-VI* is expressed at high levels in the PT cell. The path of the ISN is overlaid with a dashed line. (**H**) The lateral bidendritic neuron (LBD), a synaptic target of the transverse nerve (TN), expresses high levels of *side-VI*. (**I**) *side-VI* is expressed in the dorsal median cell (DMC, arrowheads) in the CNS. (**J**) *side-VI* expression in ventral muscles in a stage 16 embryo (dorsal and ventral borders of muscles 12 and 13 are indicated by dotted lines). (**K**) *side-VII* expression in a S14 embryo is broad in the CNS and in the trachea (outlined). (**L, M**) *side-VIII* expression at stages 15–16 in RP1, 3, 4, and five motoneurons co-stained for lim3 >Tau myc (**L**), and in the pCC interneuron co-stained for RN2-Gal4 > UAS tau-myc (**M**). (**N**) Synopsis of *side-III, IV, VI* and *VII* expression in the periphery. Anterior is left and the ventral CNS is down in all panels except C, D, I, L, M and N where anterior is up. Scale bars indicate distances.

DOI: https://doi.org/10.7554/eLife.28111.011

The following figure supplement is available for figure 6:

**Figure supplement 1.** Analysis of *side-VI* expression patterns in larvae.
DOI: https://doi.org/10.7554/eLife.28111.012

Overall, the expression patterns of the *side-III, -IV, -VI, and -VII* genes in the periphery are consistent with the idea that they could have roles like that of *side*, encoding guidance cues for motor or sensory axons. In particular, the dynamic nature of *side-VI* expression and the fact that it is expressed at intermediate and final targets of the ISN, ISNb, and TN suggests that it may play a role in guiding motor axons towards their targets through receptors expressed on these nerves. However, our phenotypic analysis (see Discussion) suggests that it has redundant functions with other guidance cues, since guidance defects at the Side-VI-expressing choice points are not observed in most segments of *side-VI* mutant embryos.

The embryonic in situ hybridization data described above show that *side-VI* is expressed in the CNS, but do not indicate whether it is transcribed in motor neurons. We thus used a *side-VI-T2A-GAL4* line (*Diao et al., 2015*) generated from a MiMIC insertion (*Venken et al., 2011*) to evaluate expression of the gene. This driver did not express in embryos, but in third instar larvae the GFP reporter driven by the GAL4 is present in all type 1b and 1 s NMJs, indicating that at this stage of development the gene is expressed in all glutamatergic motor neurons (*Figure 6—figure supplement 1*). We also observed expression in subsets of sensory neurons and in the ventral nerve cord (data not shown).

## Expression of *beat-I* and *beat-V* genes in identified motor neurons

Pipes et al. showed that expression of all *beat* genes except *beat-IIs* is restricted to the CNS at stage 16 (*Pipes et al., 2001*). We focused on five of the six *beat-V* and *beat-I* genes, for which we could readily detect expression in single cells by in situ hybridization, in order to determine whether the individual genes within these clusters had acquired unique expression patterns, as one might predict based on the fact that these duplicated genes are maintained in most or all Drosophilid species. Having shown that *side* subfamily genes are expressed in cells targeted by motor axons, we wished to determine if the genes encoding their Beat receptors were expressed in motor neurons and, if so, to identify those motor neurons. We thus performed in situ mRNA hybridization combined with simultaneous immunohistochemistry and confocal imaging using two marker lines (RN2-Gal4 and Lim3A-tau-myc [*Thor et al., 1999*; *Fujioka et al., 2003*]), to specifically identify the ISN neurons aCC and RP2, which innervate dorsal muscles (*Fujioka et al., 2003*), and the ISNb neurons RP1, 3, 4 and 5, which innervate ventrolateral muscles.

Within the *beat-I* subfamily, *beat-Ia* is expressed in both the aCC and RP2 motor neurons of the ISN (*Figure 7A,F*) where its transcription is dependent on *eve* ([*Zarin et al., 2014*] and data not shown) and in RP1, 3, 4 and 5 ([*Pipes et al., 2001*] and data not shown). *beat-Ic* is expressed in aCC and RP2, as well as in the pCC interneuron, which also expresses RN2-GAL4, but not in RP1, 3, 4,

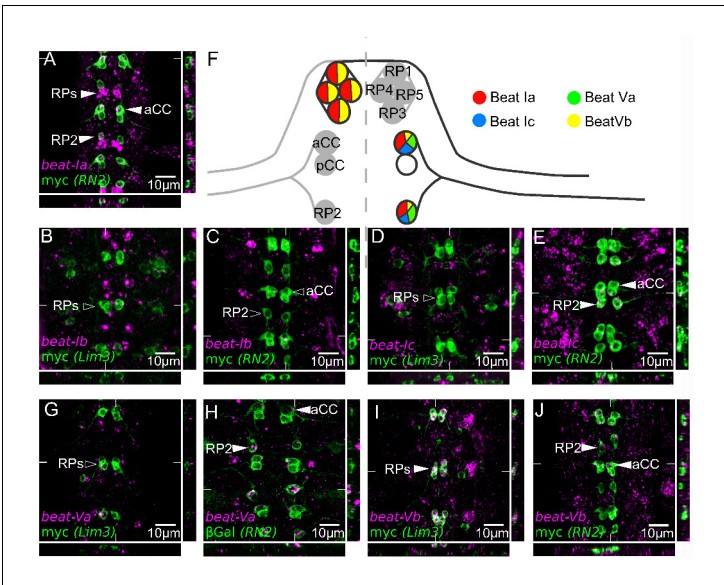

**Figure 7.** Embryonic expression of *beat-I* and *beat-V* subgroups in motor neurons indicates regulatory divergence. Fluorescent in situ mRNA hybridization for the *beat-I* and *beat-V* groups of genes (magenta). Individual motor neurons are marked (green) with anti-myc in a *Lim3A-tau-myc* (*Lim3*) line and anti-myc or anti-βGal in *RN2-Gal4 > UAS* tau-myc-eGFP and *RN2-Gal4 > UAS* LacZ (*RN2*) to reveal RP1, 3, 4, and 5 (Lim3) or aCC and RP2 (RN2) cells, respectively. (**A**) *beat-Ia* is expressed in the ISN pioneer motor neurons aCC and RP2 and in the ISNb motor neurons RP1, 3, 4, and 5 (arrowheads). (**B**) *beat-Ib* is not expressed at observable levels in the RP1, 3, 4 or 5. (**C**) *beat-Ib* is expressed only at low levels (isolated magenta dots) in aCC and RP2. (**D**) *beat-Ic* is not expressed in RP1, 3, 4 and 5, but is expressed (**E**) in aCC and RP2. (**G**) *beat-Va* is not expressed in RP1, 3, 4, or 5 motor neurons but is expressed in aCC and RP2 (**H**), clearly showing higher levels in RP2 than in aCC. (**I**) *beat-Vb* is expressed at high levels in RP1, 3, 4 and 5 motor neurons and at low levels in aCC and RP2 (**J**). All embryos are dissected to expose the CNS. Anterior is up in all images, with the ventral midline in the center. Coordinates of orthogonal slices are indicated on main panels and XY and XZ cuts are represented to the right and bottom of each panel respectively. (**F**) Expression profiles of the *beat-I* and *beat-V* genes in dorsally (aCC, RP2) and ventrally (RP1, 3, 4, 5) projecting motor neurons. Scale bars indicate distances.
DOI: https://doi.org/10.7554/eLife.28111.013

and 5 (*Figure 7D,E,F*). *beat-Ib* is expressed at low levels in aCC and pCC (dots in *Figure 6C*), but not detectably in RP1, 3, 4, and 5 (*Figure 6B*).

*beat-Va* and *beat-Vb* are differentially expressed in ISNb and ISN motor neurons. *beat-Va* is absent from RP1, 3, 4 and 5 (*Figure 6F,G*), but is expressed at high levels in RP2 and at lower levels in aCC (*Figure 6F,H*). By contrast, *beat-Vb* is expressed at high levels in RP1, 3, 4 and 5 (*Figure 6F, I*), but at lower levels in aCC and RP2 (*Figure 6F,J*). These results show that embryonic expression patterns within the CNS have diversified between clustered *beat-I* and *beat-V* paralogs.

## Discussion

We describe a new high-throughput assay for detection of protein-protein interactions, the BPIA, which employs the Bio-Plex system. In principle, this system can allow 500 unique protein-protein (bait-prey) interaction pairs to be analyzed in a single well. In our method, capture of proteins from media with streptavidin-coupled beads bypasses the purification step for bait proteins (*Figure 2*). The assay is also compatible with the use of unpurified prey proteins, thereby reducing the workload for multiplexed screenings. The small size of the beads, the ability to probe multiple interactions simultaneously, and the small volume of the binding reactions all help reduce the amount of protein and reagents needed for the assay. We were able to produce enough bait and prey proteins for the mini-interactome described here (a 23 × 23 matrix) with a single 10 cm dish transfection per protein.

As a test of the system, we used the BPIA to examine interactions between the *Drosophila* Beat and Side IgSF protein subfamilies (*Figure 1*). Beat-Ia is a receptor on motor growth cones that

recognizes Side expressed on muscles, and in the absence of Beat-Ia or Side motor axons fail to leave their bundles and arborize on their muscle targets (*Fambrough and Goodman, 1996*; *Sink et al., 2001*; *Siebert et al., 2009*; *Aberle, 2009*; *Pipes et al., 2001*). There are 14 Beat subfamily members and 8 Side subfamily members (*Figure 1*), but all of these proteins except Beat-Ia and Side itself were orphans until the global IgSF interactome revealed interactions between six other Beats and three Sides (*Özkan et al., 2013*).

In the Dpr-ome, the other IgSF network uncovered by the interactome, every Dpr protein likely to be capable of binding has an interaction partner in the DIP subfamily (*Carrillo et al., 2015*). Based on this, we predicted that there should be additional interactions to be discovered within the Beat-Side subfamily network. Using the BPIA, we were able to find three new interactions: Beat-VI::Side-II, Beat-Ic::Side-III, and Beat-Ic::Side (*Figure 3*). These results suggest that the BPIA is more sensitive than the ECIA. Like the ECIA, the BPIA should be able to find new receptor-ligand interactions even if proteins not previously known to have any interactions were tested. Of course, for both assays any candidate receptor-ligand pairs need to be confirmed as genuine using other methods. For the Beat-Side network, all three new interactions found by the BPIA, as well as the interactions between the three Beat-Vs and Side-VI found by the ECIA, were verified by SPR (*Figure 4*). We also demonstrated that Beat-Vs and Side-VI interact using cell-based binding assays (*Figure 5—figure supplement 1*) and binding to live-dissected embryos (*Figure 5*).

There are still five Beats and two Sides that remain orphans. Since the structure of Beat-Side complexes is unknown, we cannot determine whether these Beats and Sides are likely to be able to bind, but we speculate that at least the three Beat-IIIs are likely to have Side partners. The Beat-II and Beat-V clusters each interact with a single Side partner, and perhaps the Beat-IIIs interact with one of the two orphan Sides. It is possible that changes in methodology, such as using more highly multimerized preys and/or baits, could increase sensitivity and allow detection of additional interactions.

## Expression and function of sides and beats

We examined the expression patterns of *side* and *beat* genes in order to obtain insights into their possible functions. Most *sides* are expressed in cells in the periphery as well as in the CNS, while most *beats* are expressed only by CNS neurons, including motor neurons (*Figures 6* and *7*) (*Pipes et al., 2001*). Beat-Ia::Side interactions are required for normal motor axon guidance, and highly penetrant motor axon defects in which muscles remain uninnervated are observed in mutants lacking either protein (*Fambrough and Goodman, 1996*; *Sink et al., 2001*; *Siebert et al., 2009*). By contrast, partial loss of function of *beat-Ib*, *beat-Ic*, both *beat-IIs*, or *beat-VI* causes motor axon defects with less than 20% penetrance (*Pipes et al., 2001*). Genetic redundancy is a common theme in motor axon guidance (see ref. [*Zarin et al., 2014*]), so it is not surprising that only low-penetrance defects are observed when Beat paralogs are not expressed. Given that Beat-Ia and Side both interact with other partners (*Figure 3*), it is perhaps remarkable that *beat-1a* and *side* have such strong phenotypes as single mutants.

We found that Beat-V and Side-VI also have redundant functions in motor axon guidance. *side-VI* is expressed in motor axon targets, including muscles 12 and 13 (*Figure 6*) and interacts with the three Beat-Vs, at least two of which are expressed in subsets of motor neurons (*Figure 7*, *Figure 5—figure supplement 2*). Beat-V::Side-VI interactions produced the strongest signals in both the ECIA and BPIA (*Figures 2B* and *3*). We observed low-penetrance (~1/5 of stage 17 embryonic hemisegments affected) muscle 12 innervation defects in *side-VI* insertion mutants or in deletion mutants lacking all three *beat-V* genes (unpublished results). There were also low-penetrance ISN guidance defects in both mutants. The fact that most muscle 12 s are innervated normally in *beat-V* or *side-VI* mutants indicates that, while Beat-V::Side-VI interactions may contribute to correct targeting of the RP5 axon to muscle 12, other cues must also be involved. Muscles 12 and/or 13 also express Wnt-4 (a repulsive ligand) and the LRR protein Capricious (Caps; probably an adhesion molecule), and low-penetrance RP5 targeting defects are observed in *Wnt-4* (*Inaki et al., 2007*) and *caps* mutants (*Kurusu et al., 2008*). Perhaps muscle 12 is distinguished from other nearby muscles by a set of partially redundant cues, so that strong targeting phenotypes are not observed in any single mutant.

Although Beat and Side paralogs may not be central to motor axon guidance, their expression patterns suggest that they could be important for determining synaptic connections within the CNS. *side-VIII*, encoding an orphan Side, is expressed in a small subset of embryonic CNS neurons

(*Figure 6*). In the optic lobe of the pupal brain, an RNAseq analysis of two photoreceptors (R7 and R8) and five types of lamina neurons (L1-L5) revealed that *beats* and *sides* have highly specific expression patterns (*Tan et al., 2015*). For example, *beat-VII* is specific to L2, *beat-VI* to L5, *beat-IIa* to L3 (with lower levels in L4), and *beat-IIIc* to R8, being expressed at much higher levels in those cells relative to all other cells. *side* and *side-III* are specific to L3, *side-II* is specific to L1, *side-IV* is specific to L2, and *side-V* is specific to L5. R7, R8 and each of the L neuron types synapse with different sets of neurons in the medulla, a ten-layered structure that processes visual information from the retina and lamina. It has been observed that R and L neurons expressing specific Dprs often form synapses on medulla neurons expressing DIPs to which those Dprs bind in vitro (*Carrillo et al., 2015*; *Tan et al., 2015*). In a similar manner, perhaps some of the medulla neurons that are postsynaptic to L or R neurons expressing specific Sides or Beats express their in vitro binding partners, and these Beat-Side interactions might be important for synapse formation or maintenance.

## Materials and methods

### Bioinformatics and phylogenetics

Orthologs for the *beat* and *side* genes in the 12 sequenced *Drosophila* species were established using a reciprocal BLAST approach, first against the annotated predicted transcript databases (*Clark et al., 2007*). Where full length orthologous coding sequence had not been predicted in the public databases, coding sequences of the N terminal ectodomains were inferred and annotated by aligning the full length orthologs from the closest related species against the genome assembly, and other available predicted transcripts in the host. Protein domains were inferred using the online implementations of SMART (*Schultz et al., 1998*), HMMER (*Eddy, 2011*) and DOUT-finder (*Novatchkova et al., 2006*). Multiple sequence alignments were carried out using the Muscle, t-coffee (*Notredame et al., 2000*) and clustal-Ω (*Sievers et al., 2011*) algorithms. Alignments were manually edited in SeaView (*Gouy et al., 2010*) and UGENE (*Okonechnikov et al., 2012*); poorly aligning sequences were removed. Maximum likelihood protein phylogenies and bootstrap analyses were performed using RaxML source code (*Stamatakis, 2006*) and RaxML via the CIPRES Science Gateway and visualised and edited in SeaView.

We could not identify orthologs in all 12 Drosophilids for all Sides, likely due to incomplete genomic sequence rather than to stochastic loss of some non-functionalized paralogs after gene duplication as predicted by Ohno's theory (*Ohno, 1970*). The missing orthologs within Side clusters are likely due to the limitations of the methods used to identify them because: (a) Side paralog clusters containing low numbers of orthologs present similar inter-species divergence levels as those containing high numbers of orthologs, hence equal selective constraints; (b) Evolutionary instability of functionally redundant gene copies, which would lead to the non-functionalization and erosion of redundant paralogs, is not a plausible evolutionary explanation for missing orthologs since the large inter-Side divergence levels imply that paralogs diverged functionally after gene duplication, and thus were not functionally redundant, and (c) the loss of redundant paralogs is expected soon after duplication (*Lynch and Conery, 2000*), likely pre-dating speciation.

### Plasmids, cell culture and protein expression

Bait expression vectors were modified from the pECIA14 vector (*Özkan et al., 2013*). An Avitag (Avidity) was added between the hexahistidine and FLAG tags at the C-terminus of the vector with standard cloning procedures to make a new Gateway (Thermo Fisher, Waltham, MA) destination vector. ECD sequences were moved from entry vectors for Beats and Sides, described in (*Özkan et al., 2013*), into the modified pECIA14 vector using LR Clonase II (Thermo Fisher). Prey proteins were expressed from the pECIA2 vector (*Özkan et al., 2013*).

All proteins, excepting the unpurified prey, were expressed in *Drosophila* Schneider 2 cells grown in S2 media with 10% fetal bovine serum, 50 units/mL penicillin and 50 µg/mL streptomycin. The unpurified prey proteins were expressed in Sf-900 III media (Thermo Fisher). Proteins were transfected using Effectene (Qiagen, Hilden, Germany), following manufacturer's instructions. Copper (0.5 mM CuSO$_4$) was added the day after transfection to induce expression of protein. For the baits, 3 mM biotin was also added to the media to facilitate in vivo biotinylation. Prey proteins were purified using Ni-NTA resin, following standard procedures.

## Bio-Plex bead conjugation and assay

We first explored direct coupling by conjugating purified AP bait proteins to bead regions. Anti-AP antibody was added to the coupled beads, followed by PE-conjugated secondary antibody, and the bead mixture was run on the Bio-Plex machine to evaluate coupling efficiency. We found that this direct coupling method was not optimal, as different bait proteins coupled to the beads with very different efficiencies (data not shown), and a great deal of protein was lost during the purification steps.

For the affinity capture method, Bio-Plex Pro Magnetic COOH Beads (Bio-Rad, Hercules, CA) were coupled to streptavidin following the manufacturer's instructions, and beads were blocked with 1% i-Block (Tropix, Bedford, MA) in PBS. Bait protein constructs were transfected into 10 cm dishes of S2 cells, and the medium from each dish was concentrated in Amicon centrifugal filters to 1 mL. The baits were then captured directly from concentrated media with 4–8 µl of beads (corresponding to 50,000–100,000 beads per region). Purified (or unpurified) prey was added to the bead mix and incubated overnight at 4° C. For most purified prey proteins, we added 1 µg protein in a 100 µl reaction. For some preys, because of differential stickiness and low expression levels, different amounts were added. These ranged from 0.05 ug to 4 µg. For unpurified protein, 50 µl of protein in media was added to an overall reaction volume of 100 µl. The next day, beads were washed with PBST containing 0.02% i-Block and incubated with anti-V5 antibody (Invitrogen) at 2 µg/mL. The beads were then washed again and incubated with PE-conjugated goat anti-mouse IgG (Santa Cruz Biotechnology. Dallas, TX). The beads were washed again, transferred into a 96-well plate and run on the Bio-Plex 200. We tried to at use at least 1000 beads per region, but because of differential bead loss during the various incubation and wash steps, different numbers of beads were counted for each region. For our analysis, we always counted at least 35 beads per bead region. Each reaction was run in duplicate.

To test whether jumping occurs, we coupled four different bead regions to streptavidin. Three of the streptavidin-coupled bead regions were used to capture three different pentamerized, biotinylated proteins. The beads were then mixed together and incubated overnight with anti-AP antibody, followed by PE-conjugated secondary antibody, and run on the Bio-Plex. Strong PE signal was detected for the bead regions with captured bait proteins, while streptavidin beads with no bait protein had no detectable signal over background (data not shown). These results show that there is no jumping of proteins between different bead regions.

## Bio-Plex data analysis

Before generating Z scores, Özkan et al. (*Özkan et al., 2013*) used a cutoff to eliminate outliers (high values, probably due to binding), as these would artificially inflate the mean and standard deviation (SD). Since our data set is relatively small (a 23 × 23 matrix), and a large fraction of the proteins interact with each other (since we are using preselected proteins that are already known to be part of a network) we could not exclude signals due to genuine binding as outliers, as that would eliminate much of the data. By contrast, in the 202 × 202 matrix of the global interactome, the probability that any randomly chosen pair of proteins actually bind to each other is very low.

To process our data, then, we utilized bootstrapping of the median for each row and each column. We construct an N x N matrix $\mathbf{X}$ with the rows and columns containing the N proteins in the same order. The rows denote the prey and the columns denote the bait. Thus, the ith prey interaction with jth bait is quantified by $\mathbf{X}(i, j)$, and the jth prey interaction with the ith bait is quantified by $\mathbf{X}(j, i)$. We then selected with replacement N random samples from the ith column of the matrix $\mathbf{X}$. The process was repeated B times (B = 300 was used) to obtain N B-dimensional vectors. Similarly, we selected with replacement N random samples from the ith row of $\mathbf{X}$, and the process was repeated to obtain N B-dimensional vectors. The mean and standard deviation of each of the N rows and N columns were calculated and each component in $\mathbf{X}$ was Z-scored with respect to the column and row statistics to obtain two N x N matrices $\mathbf{X}_{zc}$ and $\mathbf{X}_{zr}$, respectively. A matrix $\mathbf{X}_{zrc}$ was formed via the element-by-element computation $\mathbf{X}_{zrc}(i,j) = (\mathbf{X}_{zr}(i,j)+\mathbf{X}_{zc}(i,j))/2$. In the scenario of both $\mathbf{X}_{zrc}(i,j)$ and $\mathbf{X}_{zrc}(j,i)$ being positive, the geometric mean of $\mathbf{X}_{zrc}(i,j)$ and $\mathbf{X}_{zrc}(j,i)$ were computed. If the geometric mean exceeded the threshold of five, then the i and j pair were labeled as an interaction. We were able to recapitulate all interactions found in the ECIA except for BeatIa::Side, which was clearly observed in one orientation (*Figure 2B*) but not detected in the other due to failure of

expression of Beat-Ia AP bait. This interaction was not scored because the geometric mean cannot be calculated if one of the Z scores is zero, which was the case for Beat-Ia-AP bait and Side prey. The interaction between Side-II and Side-III was very strong in one orientation but just below our cut-off in the opposite orientation, so we have preserved this interaction, seen in the original ECIA.

## Protein expression, Purification and Surface Plasmon Resonance

All Beat and Side extracellular domains with C-terminal hexahistidine tags were expressed in and secreted from *Trichoplusia ni* High Five Cells using the baculovirus system. Proteins were first purified with Ni-NTA agarose resin, followed by size exclusion chromatography using Superdex 75 or 200 10/300 columns (GE Healthcare). For capturing on Surface Plasmon Resonance chips, Side-VI (CG34114), Beat-Ic and Beat-VI expression constructs also included a biotin acceptor peptide sequence, which was biotinylated using *E. coli* BirA biotin ligase, and this allowed proteins to be captured on SA (streptavidin) Biacore chips (GE Healthcare). Beat-Va, Vb, Vc, Side-II, and Side-III were titrated in the mobile phase over the SA chips.

Side-VI, Beat-Ic, Beat-VI and Beat-Va, Vb, and Vc expression constructs included complete ectodomains. Due to problems with expression and/or purification for full-ectodomain constructs of Side, Side-II, and Side-III, shorter fragments of these Side ectodomains were used for SPR, based on the knowledge that the first IgSF domains of Sides are sufficient for Beat-Side interactions (unpublished data). The following constructs were used during SPR experiments: N-terminal two IgSF domains of Side, N-terminal single-IgSF domains of Side-II and Side-III.

Surface Plasmon Resonance (SPR) experiments for Side-VI against Beat-Va, -Vb and -Vc were performed on a Biacore T100 (GE Healthcare), and for Beat-Ic and Beat-VI against Side-II and Side-III were performed on a Biacore 3000. Unless noted, all SPR binding measurements are done in HBSp+ (GE Healthcare), which includes 10 mM HEPES pH 7.2, 150 mM NaCl, and 0.05% surfactant Polysorbate 20. To prevent non-specific binding to Biacore chip surfaces, Beat-Va and Vb binding experiments were performed with HBSp +containing 500 mM NaCl and 15% Glycerol. For similar reasons, Side-II and Side-III binding was performed in the buffer HBSp +and 1% (w/v) bovine serum albumin (BSA).

Binding between Sidestep (mobile phase) and Beat-Ic (stationary phase) could also be observed, but precipitation of Sidestep prevented us from collecting a titration series.

## Cell surface binding assays

AP-fusion constructs were generated by Gateway recombination into a destination vector, pUAS-LPGWAP, containing a metallothionein promoter N- terminal leader peptide and C-terminal AP. Secreted AP-ectodomains were produced in *Drosophila* S2 cells by co-transfecting the pUAS-prey-AP and pAct-Gal4 plasmids using FuGENE HD transfection reagent (Promega, Madison, WI). Cell surface binding assays were adapted from those previously described (*Cheng and Flanagan, 1994*). Briefly, $10^6$ *Drosophila* S2 cells were seeded in 6-well plates, transfected with cell surface bait or control constructs, expression was induced and cells were harvested by centrifugation. Cells were washed and incubated with 0.5 nM Prey-AP or LP-AP (control) conditioned S2 media for 90 min at room temperature, washed, and bound AP activity was measured.

## Live staining with Side-VI-AP in embryos expressing Beat-Vb

UAS-Beat-Vb x Tub-GAL4 or wild-type embryos (*Figure 5*) were collected, dissected, and stained following procedures described in Lee *et al* (*Lee et al., 2009*). Similar methods were used for wild-type embryos in *Figure 5—figure supplement 2*. Dissected embryos were stained with Side-VI-AP (in S2 media), followed by primary antibodies rabbit anti-AP (Serotec) and mAb 1D4. Secondary antibodies used were Alexa-Fluor 568 anti-rabbit and Alex-Fluor 488 anti-mouse (Invitrogen) at a 1:1000 dilution. Images were collected on a Zeiss LSM 710 using a 40X objective.

## Immunohistochemistry and in situ mRNA hybridization

In situ mRNA hybridization was performed as previously described (*Zarin et al., 2012*). Probes were generated from cDNA vectors (Drosophila Genomics Resource Centre; *beat-Ia* cDNA kindly provided by H. Aberle) for the genes of interest and specific motor neurons were labeled in the following stocks: *RN2-Gal4::UAS-tau-myc-GFP*, *RN2-LacZ* (*Fujioka et al., 2003*), *Lim3A-tau-myc*

(*Thor et al., 1999*). Motor axon staining to define phenotypes was done as described in *Patel (1994)*.

## Visualization of Side-VI and Beat-Va expression patterns in larvae

Third instar larvae of Side-VI and Beat-Va T2A-GAL4 lines driving GFP were dissected following procedures described in (*Menon et al., 2009*). Dissected larvae were stained with rabbit anti-GFP (Invitrogen) at 1:500, followed by rhodamine-conjugated anti-HRP (Jackson ImmunoResearch) at 1:50 and Alexa-Fluor 488 anti-rabbit (Invitrogen) at 1:1000. Samples were imaged with a Zeiss LSM 710 with a 40X objective. Images were processed with ImageJ and Adobe Photoshop.

## Acknowledgements

This work was supported by NIH grants to KZ (R37 NS28182), and to EÖ (RO1 NS097161), by the Klingenstein-Simons Fellowship in the Neurosciences to EÖ, and by SFI grants 07/IN.1/B913 and 08/RFP/NSC1617 to J-P L. We thank Aref Arzan Zarin for preliminary genetic analysis. We thank Elena Armand and Suzanne Fisher for technical assistance, Maria Prats for preparation of AP supernatants for the experiments of Figs. S2 and S3, and Violana Nesterova for figure preparation. We thank Kaushiki Menon and Namrata Bali for help with larval staining. We acknowledge Dr. Elena Solomaha and the University of Chicago BioPhysics Core Facilities for training with and access to a Biacore 3000. We thank Laura Quintana Rio, Lalanti Venkatasubramanian, and Richard Mann (Columbia) for the Side-VI-T2A-GAL4 line.

## Additional information

### Funding

| Funder | Grant reference number | Author |
|---|---|---|
| National Institutes of Health | RO1 NS097161 | Engin Özkan |
| The Klingenstein-Simons Fellowship Awards in the Neurosciences | | Engin Özkan |
| Science Foundation Ireland | 07/IN.1/B913 | Juan-Pablo Labrador |
| Science Foundation Ireland | 08/RFP/NSC1617 | Juan-Pablo Labrador |
| National Institutes of Health | R37 NS28182 | Kai Zinn |

The funders had no role in study design, data collection and interpretation, or the decision to submit the work for publication.

### Author contributions

Hanqing Li, Conceptualization, Resources, Data curation, Formal analysis, Validation, Investigation, Writing—original draft, Writing—review and editing; Ash Watson, Hyung-Kook (Peter) Lee, Investigation, Visualization; Agnieszka Olechwier, Michael Anaya, Validation, Investigation, Methodology; Siamak K Sorooshyari, Software, Investigation, Methodology; Dermott P Harnett, Investigation, Methodology; Jost Vielmetter, Resources, Data curation, Funding acquisition, Validation, Investigation, Methodology, Project administration; Mario A Fares, Conceptualization, Software, Investigation, Methodology; K Christopher Garcia, Conceptualization, Resources, Project administration; Engin Özkan, Conceptualization, Data curation, Formal analysis, Supervision, Funding acquisition, Validation, Investigation, Visualization, Methodology, Project administration, Writing—review and editing; Juan-Pablo Labrador, Conceptualization, Resources, Data curation, Formal analysis, Supervision, Funding acquisition, Validation, Investigation, Visualization, Methodology, Project administration, Writing—review and editing; Kai Zinn, Conceptualization, Data curation, Formal analysis, Supervision, Funding acquisition, Investigation, Visualization, Methodology, Writing—original draft, Project administration, Writing—review and editing

## Author ORCIDs

Siamak K Sorooshyari [ID] http://orcid.org/0000-0002-1172-6291
K Christopher Garcia [ID] http://orcid.org/0000-0001-9273-0278
Engin Özkan [ID] http://orcid.org/0000-0002-0263-6729
Kai Zinn [ID] http://orcid.org/0000-0002-6706-5605

## Decision letter and Author response

Decision letter https://doi.org/10.7554/eLife.28111.015
Author response https://doi.org/10.7554/eLife.28111.016

## Additional files

**Supplementary files**
• Transparent reporting form
DOI: https://doi.org/10.7554/eLife.28111.014

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
