## [Decision Letter]

Thank you for submitting your article "Deconstruction of the Beaten Path-Sidestep interaction network provides insights into neuromuscular system development" for consideration by *eLife*. Your article has been favorably evaluated by Eve Marder (Senior Editor) and three reviewers, one of whom, Graeme W Davis (Reviewer #1), is a member of our Board of Reviewing Editors. The following individual involved in review of your submission has agreed to reveal their identity: Sean Sweeney (Reviewer #2).

Three reviewers have read and discussed the manuscript. The main achievement of this manuscript is describing a new method, Bio-Plex Interactome Assay (BPIA) for high-throughput detection of interactions between extracellular and secreted candidate proteins. Although the method is not entirely novel, it has been significantly improved by the authors, in terms of efficiency and signal-to-noise ratio.

The authors use the BPIA method to successfully and convincingly identify novel physical interactions between several uncharacterized members of the Beat and Side extracellular protein family members. They then provide mostly descriptive data regarding the expression patterns of these novel Beat and Side proteins in *Drosophila* embryos and larvae. Finally, the authors provide evidence for a functional role of a particular Beat and Side interaction (BeatV and SideVI).

All three reviewers agree that the biochemical data regarding the dissection of Beat and Side protein interactions is an advance that would be well suited to publication in *eLife*. However, all three reviewers also felt that the manuscript ended on a relatively weaker note with the functional/ genetic analysis of BeatV and SideVI. Missing in this analysis were some standard genetic tests for specificity including genetic rescue and a test for trans-heterozygous interaction. This aspect of the manuscript could be significantly strengthened to match the enthusiasms for the biochemical approach and advance.

The reviewers, in consultation with a Senior Editor at *eLife*, agree that the authors could choose between two approaches to move forward:

1) The authors may feel that strengthening their genetic analysis is straightforward and could be accomplished in a rapid time frame.

2) Alternatively, the authors could remove the final functional analysis of growth cone guidance from the paper and publish the work solely based on the biochemical elucidation of Beat and Side interactions. If this route were chosen, the authors could then take advantage of the option that *eLife* offers, to publish an 'Advance', which is a follow-up study that builds upon an existing *eLife* paper. Taking this route, the authors could allow themselves additional time to fully develop the functional analyses without holding up their current biochemical work.

The choice is up to the authors about which approach they would find most satisfactory. Obviously, if you already have in hand most of what you would need for option 1, that might be advisable. If the additional work would take more than 6 weeks, then clearly option 2 is preferable.

When considering revisions, the following comments should be taken into account. However, if the authors decide to focus their work solely on the biochemical analyses, they do not, obviously, need to address comments on the functional interaction studies. These comments can serve as a guide to a future Advance publication.

Major Concerns:

1) The side-VI LOF experiment is statistically significant, so could a rescue not be attempted? A double heterozygote analysis? (the side-VI MiMiC allele against the beat Va,b,c Df?). Alternatively, some of the sides and beats are proposed to be secreted, could mis-expression of some of these proteins (or some of the other sides and beats, minus their TM domain) be used to prove interaction in vivo?

2) I would like to see some improvement in the explicatory power of the diagrams and matching figure legends (the main text explains the procedures well, the figures not so much), and scale bars on the images (I don't think I see one throughout).

3) The heavy use of professional, field-specific jargon, in Figure 6 and Figure 7, in particular, makes those sections of the paper less accessible to an average *eLife* reader.

4) The axon guidance phenotypes of Beat-V and Side-VI mutants are relatively weak. The authors mention they don't know if the MiMic insertion mutants of Side are null. Have the authors tested (for example by RT-PCR or qPCR) whether these mutants are actually null? Also, have they tried knocking down Side specifically in the muscle using RNAi? Also, it could be nice to see rescue of the Side-VI mutant phenotype by expressing Side-VI in muscles.

---

## [Author Response]

*[…] When considering revisions, the following comments should be taken into account. However, if the authors decide to focus their work solely on the biochemical analyses, they do not, obviously, need to address comments on the functional interaction studies. These comments can serve as a guide to a future Advance publication.*

We were given two options for this manuscript: do further experiments to expand on the genetic analysis, and resubmit a revised version as a new paper (since such experiments would require several months to complete), or remove the phenotypic data and publish the paper without these data. We chose the second option.

The changes requested by the reviewers were quite straight forward, since most of their criticisms concerned the preliminary nature of the phenotypic analysis and we have removed those data.

The remaining concerns (not related to the phenotypic data) and our responses are detailed below:

*Major Concerns:*

*1) I would like to see some improvement in the explicatory power of the diagrams and matching figure legends (the main text explains the procedures well, the figures not so much), and scale bars on the images (I don't think I see one throughout).*

This has been done. We have modified Figure 2, adding several new boxes and more explanation of the steps in the BPIA method. The figure legend has also been expanded.

We have added scale bars to all the relevant figures.

*3) The heavy use of professional, field-specific jargon, in Figure 6 and Figure 7, in particular, makes those sections of the paper less accessible to an average eLife reader.*

It is unfortunately necessary to use many *Drosophila*-specific terms to describe the expression patterns of the beat and side genes. However, we have added summary statements to each of the sections in Results describing the overall nature of the expression patterns, so that reviewers not interested in the details can skip over the remainder of these paragraphs.